# Highly Fluorescent Green Carbon Dots as a Fluorescent Probe for Detecting Mineral Water pH

**DOI:** 10.3390/s19173801

**Published:** 2019-09-03

**Authors:** Tingyu Wang, Guoqing Chen, Lei Li, Yamin Wu

**Affiliations:** 1School of Science, Jiangnan University, Wuxi 214122, China; 2School of Science, Jiangsu Provincial Research Center of Light Industrial Optoelectronic Engineering and Technology, Wuxi 214122, China

**Keywords:** carbon dots, solvothermal, 3,5-diaminobenzoic acid, pH, fluorescent probe

## Abstract

In this report, high-brightness green carbon dots were successfully prepared using 3,5-diaminobenzoic acid as the sole precursor and synthesized in one step using a solvothermal strategy. Under the excitation of 365 nm ultraviolet light, the quantum yield of carbon dots is as high as 53.8%. Experiments revealed that the carbon dots are highly carbonized and the surface is rich in amino and carboxyl groups. The synthesized carbon dots have good water solubility, and are resistant to ions and temperature. The fluorescence intensity of CDs is sensitive to pH changes and is linearly correlated with the pH in the near-neutral range (pH = 6.0 to 9.0). Our experiments showed that carbon dots were sensitive and accurate fluorescent probes for measuring the pH value of drinking water, which could provide an effective method for measuring the pH value of water in the future.

## 1. Introduction

Carbon dots (CDs) are a new type of zero-dimension photoluminescent nanomaterial with a size less than 10 nm [1]. CDs exhibit outstanding wide-range fluorescence, stable optical properties, and biocompatibility. They also have a simple and abundant synthetic path and pollution-free precursors. CDs have been found to have a series of interesting properties, researched by many scholars in recent years. For example, some unique optical properties such as long-wavelength emission can be applied to bioimaging [2,3,4]. Excitation-dependence is expected to play a role in multicolor imaging or LEDs [5,6]. In addition, CDs are considered to be detectable in response to some metal ions (Fe, Ag, and Hg) or pH due to their rich functional groups on their surface [7,8,9,10]. Therefore, CDs can be used as fluorescent probes for the detection of trace elements or pH in the environment or human body. In general, synthetic routes for preparing CDs are divided into two methods: bottom-up and top-down [11,12]. Compared to other synthetic routes for preparing CDs, the solvothermal method is an effective bottom-up method for synthesizing CDs that can be modified by polymerization and carbonization of precursors [13]. Thank to their excellent physical and chemical properties, there is no denying that CDs are promising nanomaterials to replace the traditional quantum dot in many fields. However, CDs’ research and application face several challenges due to their complex chemical structures, random surface functional groups, and low quantum yield (QY).

pH is an important parameter for living organisms. Biological activities can only be carried out within a limited pH range. A slight change of pH in the environment may have an effect on the organism. Therefore, it is important to observe changes in pH in the environment or in a living body in real time. As a new type of nanoluminescent material, many scholars have found that CDs’ rich fluorophores make them very sensitive to pH changes. At the same time, they have the advantages of fast response, high QY, and good visual recognition [14,15]. Many amino-rich organic materials are well suited for the synthesis of pH-sensitive CDs as precursors. For example, p-phenylenediamine and its isomers (m-phenylenediamine, o-phenylenediamine) are well suited for the synthesis of high-intensity, long-wavelength-emitting, pH-sensitive CDs. Jiao and her team chose p-phenylenediamine as the precursor and synthesized orange fluorescent nitrogen-doped CDs using hydrothermal methods. As the pH of the CDs increases, there is significant fluorescence enhancement in the pH range 1–8. It is therefore used as a biosensing platform to monitor fluctuations in the pH of living cells [16]. Precursors such as diethylamine and ethylenediamine are suitable for the synthesis of CDs with surface modification. Nie and co-workers used chloroform (CHCl_3_) and diethylamine (DEA) as synthetic raw materials to synthesize continuously adjustable full-color emission CDs by the solvothermal method by changing the reaction conditions. Based on its unique emission characteristics, a ratio pH sensor was constructed and applied to the measurement of intracellular pH values and cancer diagnosis [17]. However, most pH-dependent CDs have a large influence on the excitation wavelength [18,19], have a weak anti-ion interference ability [20,21], are easily affected by temperature changes [22], or are insufficiently sensitive to fluorescence changes in the near-neutral range of pH. This limits the application of CDs in the field of pH sensors. Therefore, it has been very valuable to develop pH-sensitive CDs with a simple synthetic route, high efficiency, and long-wavelength luminescence. Meanwhile, due to the different types of ions contained in the soil of different water sources, the pH value of drinking/mineral water will also change. This means that pH is an important parameter for evaluating the quality of drinking/mineral water. The current use of a pH meter is a common method for detecting the pH of water. Compared to pH meters, pH sensors based on fluorescent nanomaterials have higher sensitivity and can also be used for the detection of trace samples [23,24]. 

Herein, we synthesize high-quantum-yield green CDs with the sole precursor 3,5-diaminobenzoic acid by solvothermal one-pot synthesis (Scheme 1). After purification by silica gel column chromatography, the CDs were obtained. The prepared CDs have a high QY of 53.8% and are highly sensitive to pH, especially in the neutral pH ranges (pH = 6.0–9.0). Based on this feature, we use the CDs as a pH nanosensor for detecting the pH of commercially available mineral water. Experiments have shown that the CDs have reliable sensitivity as a pH sensor and can be used for small samples instead of traditional pH meters.

## 2. Materials and Methods

3,5-diaminobenzoic acid was purchased from Aladdin Industrial Corporation. Ethanol (≥99.7%), methanol (≥99.8%), dichloromethane (≥99.5%), and silica gel (Acuity FCP. 300–400 mesh) were obtained from Sinopharm Group Chemical Reagent Co., Ltd., Shanghai, China. All reagents were analytical reagent grade. BaCl_2_, CaCl_2_, CrCl_2_, CuCl_2_, FeCl_2_, FeCl_3_, KCl, MgCl_2_, MnCl_2_, NaCl, PbCl_2_, NaOH, HCl, NaH_2_PO_4_, Na_2_HPO_4_, citric acid, sodium citrate dihydrate (C_6_H_5_Na_3_O_7_·2H_2_O), and Tris were all purchased from Sinopharm Group Chemical Reagent Co., Ltd. Deionized (DI) water was used throughout the experiments. All aqueous solutions were prepared with ultrapure water (≥18.2 MΩ·cm) from a Labonova Smart system (Think-lab, Germany).

### 2.1. Characterizations

Transmission electron microscopy (TEM) image was gained from a JEOL (Peabody, MA, USA) 2100F microscope operating at a maximum acceleration voltage of 200 kV. Fourier transform infrared (FTIR) spectra were acquired on a Nicolet iS5 spectrometer (Thermo Fisher Scientific, Waltham, MA, USA) using KBr disks in the wavenumber range of 4000~400 cm^−1^. The X-ray photoelectron spectra (XPS) were recorded on an ESCALAB (Waltham, MA, USA) 250xi X-ray photoelectron spectrometer. To describe the CDs’ optical characteristics, the UV-Vis absorption spectra were measured on an 2048x14-USB2 fiber optic spectrometer (Aventes, Hemmingen, Germany). Fluorescence emission spectra and excitation spectra were recorded on FLS920 (Edinburgh, Livingston, England). 

### 2.2. Preparation of CDs and pH Buffer Solutions

The CDs was prepared using a one-step hydrothermal process. Firstly, 0.4 g 3,5-diaminobenzoic acid was dissolved in 40 mL ethanol. Then, the mixture was moved to a 100-mL tetrafluoroethylene-lined autoclave and heated at 180 °C in an electric oven for 12 h. After cooling to room temperature, the dark brown reaction was filtered and concentrated in vacuo to one-quarter of the original volume. The purification of the CDs was carried out by column chromatography. The stationary phase was silica gel and the mobile phase was a mixture of methanol and dichloromethane in a ratio of 4:1. Finally the purified CDs were freeze-dried into a powder (Appendix A) using a lyophilizer and stored in a refrigerator.

A series of pH buffer solutions were prepared at 0.1 mol/L. pH 2–6 solutions were prepared using citric acid and sodium citrate; pH = 7–8 solutions were prepared using tris and hydrochloric acid; pH = 9–13 solutions were prepared using sodium hydride and sodium hydrogen phosphate, adjusted by adding different quantities of NaOH solution. All pH buffer solutions were prepared on the same day.

### 2.3. QY Measurements

The QY of the CDs in water was determined by the reference method. The reference material is quinine sulfate (QY = 55% in 0.1 M H_2_SO_4_) and the formula is as follows [4]:$$\Phi_{x}=\frac{A_{s}}{I_{s}}\cdot \frac{I_{x}}{A_{x}}\cdot \frac{n_{x}^{2}}{n_{s}^{2}}\cdot \Phi_{s},$$
where $\Phi_{x}$ is the QY of the CDs, *A* is the absorbance of the CDs, and *I* is the integral intensity (excited at 365 nm). n is the refractive index (1.33 in water) and the subscript “*s*” represents quinine sulfate. In order to make the results more accurate, we prepared a series of different concentrations of the CDs and reference solution (adjusting the concentration so that their absorbance is within 0–0.2).

### 2.4. Method for Detecting the pH Value of Mineral Water

There were eight kinds of mineral water samples used in the experiment: Ganten mineral water, NongFu mineral water (suitable for infants), NongFu mineral water, Alps mineral water, LIFELONG mineral water, Evergrande Spring low sodium mineral water, Perrier gassy water and Wahaha soda water. First 0.5 mL of a diluted aqueous solution of CDs was added to a 5 mL mineral water sample and buffer solution with pH = 6 and pH = 8 (for emission intensity correction). Then this was shaken well and we measured the fluorescence spectrum of the solution using FLS920. The excitation wavelength was 390 nm and the emission range was from 400 nm to 700 nm. The obtained integrated intensity was taken into the pH change fitting curve to calculate the sample pH value. 

## 3. Results

### 3.1. Characterizations of the CDs

The distribution and size of the CDs can be observed from the TEM image (Figure 1a). It can be seen that the CDs are globular and well-dispersed. By counting the size of the CDs, it was found that the diameters of the CDs are within the range of 5–11 nm and the average diameter is 7.58 nm. Compared with other CDs, the CDs we prepared are larger, indicating that they have a higher degree of carbonization. The FTIR spectrum can identify functional groups on the surface of the CDs to evaluate the surface function. Figure 1b depicts the FTIR spectrum of the CDs. The obvious absorption bands at 3362 and 1383 cm^−1^ are consistent with the stretching vibration of the -NH group, which indicated that the surface of the CDs contains a large amount of amino groups. Studies have shown that more amino groups are favorable for the fluorescence emission of CDs, which is the reason for the high QY of the CDs [25,26]. The absorption bands at 3451 and 1466 cm^−1^ belong to -OH and -C=O, confirming the presence of carboxyl groups, while 1614 cm^−1^ is from -C=N in the phenazine structure. To characterize the elemental composition of the CDs, X-ray photoelectron spectroscopy (XPS) was used as in Figure 1c. Three dominant peaks at 531.2 (O1s), 399.4 (N1s), and 285.0 eV (C1s), which are composed of O1s, N1s, and C1s, were observed [5,27]. The O, N, and C atom ratio was 15.7%, 6.3%, and 78.0%, respectively. The high-resolution XPS spectrum of C1s (Figure 1d) exhibits three characteristic peaks of C=C/C-C, C-N/C-O, and COOH with binding energies at about 284.7, 285.1, and 288.2 eV, respectively. In Figure 1e, N1s contains three peaks of 398.9, 399.4, and 400.4 eV, indicating the presence of pyridine-N, amino-N, and pyrrole-N structures. The high-resolution XPS spectra of O1s shown in Figure 1f indicate that the O1s could be resolved into two peaks of 531.2 eV (C=O) and 532.4 eV (C-O).

### 3.2. The Optical Properties of the CDs

The optical properties, including the UV-Vis absorption, fluorescence excitation, and emission of the CDs, are investigated fully. It can be seen from the UV-visible absorption spectrum that the CDs have multiple electronic absorption transitions and there are three absorption peaks in the absorption spectra of the CDs (Appendix A). There are strong absorption bands at 229 nm and 260 nm, which can be attributed to the vibration of π-π* (aromatic C=C) and n–π (C-N) transitions [28,29]. This means that a large π bond is formed between the molecules of the CDs. The wide absorption band at 360 nm is caused by n-π* transition due to the C=O bond on the surface of the CDs [30], which describe the presence of carboxyl groups and amino substituents. The excitation spectrum of the CDs is similar to the absorption spectrum (Figure 2a), with two excitation peaks at 264 nm and 390 nm, and the fluorescence emission reaches a maximum at 390 nm excitation. An excitation peak close to the peak position of the absorption spectrum means that the green emission of the CDs is caused by the carboxyl group and/or C-N structure of the surface of the CDs. Figure 2b displays the fluorescence emission spectra of the CDs excited by different excitation wavelengths. It emerged that there was an unvaried emission at 493 nm under the varied excitations. This observation also implies that the PL emission of CDs is caused by one state of emission, which also proves the uniform size of the CDs [31,32]. The purified CDs have a high quantum yield of 53.8%, and the measurement methods and results are shown in Appendix A. It can be seen that the CDs have good QY in water solvents at a neutral pH. The high QY allows the CDs to be applied as a light sensor that can be recognized by the naked eye. 

### 3.3. Fluorescent Response of the CDs to pH

pH plays a critical role in biological systems, and small changes in pH in the environment can be fatal to living organisms. In the experiments, we studied the response of the CDs to pH by observing changes in the fluorescence emission spectra in different pH buffer solvents. As shown in Figure 3a, the fluorescence intensity of the CDs is stronger under acidic conditions, but sharply reduced under alkaline conditions. The strong response of the CDs to pH is considered to be derived from the protonation–deprotonation of the amino and carboxyl groups on their surface. As shown in the inset of Figure 3a, the fluorescence intensity of the CDs decreases with the increase of pH at 390 nm excitation from 4.5 to 12. The change trend can be fitted with the Boltzmann equation: I = 2626920 + 90256180/(1 + exp((pH − 7.3)/0.49)) (*R*^2^ = 0.999, pKa = 7.3). Appendix A shows that a linear relationship between the photoluminescence (PL) intensity and pH was observed in the near-neutral pH range (pH = 6.0–9.0, *R*^2^ = 0.985). When using concentrated sodium hydroxide and a hydrochloric acid solution to repeatedly adjust the pH of CDs and test their fluorescence intensity, we found that CDs are provided with good reversibility, which means that their PL behavior was reversible and sensitive (Figure 3b). Meanwhile, the response of different concentrations of CDs (expressed in absorbance) to pH was also tested (in Appendix A). In the range of 0.05 to 0.24 in absorbance (at a wavelength of 390 nm), the CDs have a good response to pH. This means that the prepared CDs can be used as a probe to detect the pH of the sample to be tested even in a small amount. The CDs’ PL intensity attenuated sharply, which explains how the CDs can be used as a sensitive pH sensor in environmental and biological systems.

### 3.4. Resistance to Temperature and Ion Interference and Luminescence Stability 

In order to show the feasibility of CDs in practical applications, fluorescence intensity of CDs at different temperatures and the anti-ion interference capability of the CDs was tested (Ex = 390 nm). Herein, the CDs were placed in an environment of 10–50 °C to measure the fluorescence intensity. Figure 3c shows that CDs have only slight fluorescence changes at 10–50 °C. Meanwhile, we found that the fluorescence intensity of CDs did not change significantly as the concentration of the NaCl solution increased. This means that the luminescence of CDs is basically not affected by the external environment, so they are ideal as detectors for pH detection as a sensor. The test of luminescence stability was displayed in Appendix A. In the light stability test experiment, CDs were exposed under a 600 W fluorescent lamp for 0–16 h to probe their luminescence stability. The experiments proved that the CDs are a luminescent nanomaterial with favorable stability. At the same time, the response of CDs to common metal ions (Ba^2+^, Ca^2+^, Cr^2+^, Cu^2+^, Fe^2+^, Fe^3+^, K^+^, Na^+^, Mg^2+^, Mn^2+^, Pb^2+^) has also been investigated. It can be concluded from Figure 3d that the fluorescence intensity of the CDs does not substantially change in the aqueous solutions of metal ions of 0.5 mM concentration, except for a slight response to iron ions, which can be attributed to the interactions between iron ions and amino or carboxyl groups on the surface of CDs [25,33]. So we can rule out the interference of typical ions in the practical applications of CDs such as cell imaging and water detection.

### 3.5. pH Detection in Mineral Water Samples

In order to verify the feasibility of the application of the fluorescent CDs pH sensor in actual samples, we selected some brands of mineral water as test samples, using the commonly-used pH meter and a CDs sensor for pH testing. The test results are shown in Table 1. It can be seen that the pH values measured by the CDs sensor are basically consistent with the pH value measured by the pH meter, which proves the feasibility of using CDs as a fluorescent pH sensor. At the same time, as zero-dimensional nanomaterials, CDs can play a role even in the detection of trace amounts of samples.

## 4. Conclusions

In summary, we successfully used a simple strategy to synthesize CDs in one step. The prepared CDs emit a bright green light with an average size of 7.58 nm. It has been shown that the surface is rich in carboxyl and amino groups. The CDs have many excellent optical features such as good water solubility, up to 53.8% QY, good light stability, and anti-ion interference capability. At the same time, we found that the CDs have a sensitive response to pH changes, especially in the near-neutral pH range. Therefore, we designed a simple method for testing the pH of mineral water using the CDs as a probe. Studies have shown that CDs have high sensitivity as pH sensors and are expected to be applied in the future.

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
