# Peer review of "Highly Fluorescent Green Carbon Dots as a Fluorescent Probe for Detecting Mineral Water pH"

_sensors, 2019, doi:10.3390/s19173801_

Round 1
Reviewer 1 Report
In this work authors have synthesised in one step fluorescent carbon dots (CDs) with emission in the green, using 5-diaminobenzoic acid as carbon source. CDs have been characterized, showing good water solubility and high fluorescence quantum yield which depends strongly on the pH of the solution, especially in the near-neutral pH range. The nanoparticles have been used as a sensor to determine with precision the pH value of drinking water samples.
I consider that the work is of interest and the study is well organized. The physical measurements presented are of acceptable quality and the manuscript is generally well written. I think that the idea behind the work is interesting but the application is very limited. In my opinion, there is some point that authors must clarify before paper is ready for publication:
Specific points:
1. Extinction coefficient of the CDs at 390 nm should be provided. Which is the range of CDs concentration that can be used for the pH determination?.
2. Author claims that a linear relationship between fluorescence intensity and pH is observed in the near-neutral pH range (6-8.5). This linear plot should be provided, with error bars, in order to determine the sensitivity of the detection method. Authors have to demonstrate that it is possible to obtain the pH value with two decimals, as in Table 1.
3. Are the changes in fluorescence reversible? Can the same CDs be used more than once?
4. I consider that, in addition to pH, there are certain parameters that could modify the fluorescence of CDs, such as temperature or changes in ionic strength, giving rise to anomalous pH values. Have authors explored these effects?
5. Currently, there are many works in literature, showing the synthesis and characterization of carbon quantum dots ant their use as pH sensor. Authors do not compare their results with those obtained for other authors. In fact, I consider that the number of references in the paper is small relative to the published literature on this subject.
6. Which is the advantage of using this sensor to determine pH in drinking mineral water, instead of the use of pHmeter, pH-dependent fluorophores or pH-dependent chromophores (indicators)?.
7. What do authors think about the possibility of use these CDs for determination of pH in drinks with color? I think that it could be a problem because many of these drinks absorb in the same spectral range that CDs or where they emit fluorescence, inducing artifacts in the pH determination.
8. What do authors think about the possibility of use these CDs in biological assays for determining intracellular pH?
9. In conclusion, authors have to convince me of the importance of these new CDs comparing to those reported in literatura
Reviewer 2 Report
Dear Editor,
There are lots of carbon dots (CDs), which are seldom sensitive to pH changes individually but mostly to some molecules, temperature, or moisture simultaneously. However, none of these papers was introduced in the Introduction section and compared with this study. I do not suggest it could be published by the journal, Sensors, as it lacks novelty.
Reviewer 3 Report
In this paper the author reported the preparation of carbon dots using 3, 5-diaminobenzoic acid as the sole precursor and synthesized in one step using solvothermal strategy. The photo-physical properties of this type of CDs was also characterized. The research itself is a full story and the material (reaction) used in this study is novel for the preparation of CDs. The presentation of this paper is well organized. Thus, from these points I am not against the publication of this paper. I only asked the author to add in some more introduction of literature summary of using 3, 5-diaminobenzoic acid kind of aromatic amino/acid analogues for the preparation of CDs.Author Response
Please see the attachment

Round 2
Reviewer 2 Report
I still do not suggest this submission to be published in Sensors as it lacks novelty. Using a chemical compound as the carbonaceous materials for making carbon dots (CDs) is not novel because there are millions of organic acids which have not been used for making CDs. The high quantum yield of the studied CDs is not related to their sensitivity to the pH changes. Furthermore, the studied CDs could only be measured in a narrow range, pH 6.0~9.0. Application of a pH sensor in drinking mineral waters is common. Tests in different ions and ionic strength are also regular processes. Although the fluorescence of CDs was stable at 10~50 oC, there was no further test on different pH values at 10~50 oC and application in hot or cold mineral waters.
